# Incorporation of ^4^*J*-HMBC and NOE Data into Computer-Assisted Structure Elucidation with WebCocon

**DOI:** 10.3390/molecules26164846

**Published:** 2021-08-11

**Authors:** Matthias Köck, Thomas Lindel, Jochen Junker

**Affiliations:** 1Alfred Wegener Institute, Helmholtz Centre for Polar and Marine Research, 27570 Bremerhaven, Germany; 2Institute of Organic Chemistry, Technical University of Braunschweig, 38106 Braunschweig, Germany; th.lindel@tu-bs.de; 3Oswaldo Cruz Foundation–CDTS, Rio de Janeiro 21040-900, Brazil

**Keywords:** NMR, structure elucidation, HMBC, NOE, CASE, web-based tools

## Abstract

Over the past decades, different software programs have been developed for the Computer-Assisted Structure Elucidation (CASE) with NMR data using with various approaches. WebCocon is one of them that has been continuously improved over the past 20 years. Here, we present the inclusion of ${}^{4}J$_CH_ correlations (${}^{4}J$-HMBC) in the HMBC interpretation of Cocon and NOE data in WebCocon. The ${}^{4}J$-HMBC data is used during the structure generation process, while the NOE data is used in post-processing of the results. The marine natural product oxocyclostylidol was selected to demonstrate WebCocon’s enhanced HMBC data processing capabilities. A systematic study of the ${}^{4}J$_CH_ correlations of oxocyclostylidol was performed. The application of NOEs in CASE is demonstrated using the NOE correlations of the diterpene pyrone asperginol A known from the literature. As a result, we obtained a conformation that corresponds very well to the existing X-ray structure.

## 1. Introduction

Together with mass spectrometry, one- and two-dimensional NMR experiments constitute the backbone of structure elucidation of unknown compounds in Organic Chemistry. Following the identification of hydrogen-carbon and hydrogen-nitrogen bonds in the HSQC-based suites of experiments, ${}^{1}$H,${}^{13}$C- and ${}^{1}$H,${}^{15}$N-HMBC-derived connectivity data will allow to propose the constitution of a new compound. As a key problem, the translation of HMBC correlations to geometrical bond distances is ambiguous, leaving the possibility of two to more than four bonds between the correlating partners. The intensity of an HMBC peak will not always exclude its interpretation as a long-range correlation (more than three bonds).

Over the decades, many different methods have been implemented, the most prominent being fragment assemblers [1,2,3,4,5,6], expert systems [7,8,9], structure generation by reduction [10], logic engines [11], stochastic structure generators [12], combinatorial brute force [13,14,15,16,17], databases of ${}^{13}$C NMR chemical shifts and fragments [18,19], combinatorial structure generation with restraints [20,21], genetic algorithms [22,23], simulated annealing [24], convergent structure generation [25,26], evolutionary algorithm [27], fuzzy structure generation [28], and expert systems with DFT [29]. However, CASE remains a challenge [29,30,31,32,33,34]. The basic issue is that the relation between a small molecule and its NMR correlation data is not reciprocal. If one back-calculates the common NMR correlation data (COSY, HMBC, and 1,1-ADEQUATE) for a specific molecule and then use this theoretical correlation data set to calculate the structure, we might obtain more than one solution. A change in the experimental conditions, such as using a different solvent, might increase the number of observable correlations [35], but also requires more NMR measurement time. Hence, trying to make better use of existing data would be preferred. Many experimental data sets contain ${}^{4}J$-HMBC correlations. However, so far, these correlations are excluded from the computational analysis, as almost all NMR-based structure generators interpret HMBC correlations as relations over two or three bonds. Considering that reliable identification of ${}^{4}J$-HMBC correlations can be difficult and that as many data as possible should be used for a complete and comprehensive CASE investigation, ${}^{4}J$-HMBC correlations should be included in the HMBC data interpretation.

WebCocon is a web service implemented as a two-stage process for structural elucidation based on NMR correlation data (see Figure 1). The first stage uses a WWW interface for the generation of the input file for Cocon. The data for the input file can be inserted manually, taken from an existing input file, or taken from a NMReDATA file. As a very helpful feature when checking a structural proposal, theoretical data can be generated from an existing molecule. The input file is then submitted to the server for the generation of structural proposals using Cocon [20,36,37,38]. Originally, Cocon accepted COSY, ${}^{2}J$_CH_ and ${}^{3}J$_CH_ HMBC, NHMBC [35,39], and 1,1–ADEQUATE [20,35,36,38,40] correlation data. Now, any HMBC correlation also can be interpreted as ${}^{4}J$_CH_ [41]. In order to limit the impact on the number of generated structures, a parameter called “4J-Flag” keeps track of how many correlations are interpreted as ${}^{4}J$-HMBC, and the maximum value for this parameter can be limited by the user. Setting this parameter to zero means that no ${}^{4}J$_CH_ interpretation of the HMBC data is allowed, setting it to –1 means that any number of HMBC correlations can be interpreted as ${}^{4}J$_CH_ correlation. Any other value defines the maximum number of HMBC correlations that can be interpreted as ${}^{4}J$_CH_.

WebCocon’s second stage prepares the results of the first stage for visualization on the client. Originally, the constitutions were presented as 2D drawings of the molecules without any particular order. This stage was later improved by the implementation of the statistical filter [42], where post-processing is based on a molecular dynamics (MD) calculation. Proposed constitutions, for which the MD can not create parameter sets are put at the end of the proposals list. All other proposals are ranked by their force field total energy and presented starting with the lowest energy. This processing uses smi23D [43], a freely available MD software. The processing is fast and improbable structures are reliably flagged as such, but no minimization parameters are available and restraints cannot be defined. Further processing methods have now been implemented on the server. A more capable molecular dynamics calculation is now available based on OpenBabel v3.1.0 [44]. It produces minimized structures with lower total energy but at the cost of a higher calculation time. The run time for the post-processing with MD is optimized by identifying different assignments resulting in identical constitutions using canonical SMILES [45,46], such that only one conformation is determined for each of them.

Although NOEs do not encode connectivity between atoms directly, they require that the constitution of a given molecule can assume a conformation that allows their fulfillment. This is frequently used in publications to justify a choice of constitution and configuration, as a possible resulting conformation would allow the observed NOEs to be fulfilled, but rarely is this argument backed up by molecular modeling. The integration of NOEs as restraints in the post-processing of suggested constitutions using restrained molecular dynamics (MD) or distance geometry (DG) will achieve the same effect by ranking conformations that fulfill the NOEs better are now backed via molecular modeling. WebCocon allows for the specification of NOEs together with the correlation data. However, as hydrogen atoms currently are only handled implicitly, NOEs to protons from CH${}_{2}$ groups are defined as being in an average position based on the proton’s positions. With this approach, diastereotopic protons currently cannot be differentiated and stereochemistry cannot be determined. Additionally, the assignment of NOE bearing atoms to different positions in the the constitution becomes important, as this might change the NOE involved. Therefore, when using NOEs, conformations have to be calculated for all assignments of all constitutions in order to identify the best solution.

The generation of 3D coordinates from connectivity information using MD normally is performed by a fragment-based construction of an initial conformation that is then optimized by the MD. This approach, as implemented by OpenBabel and smi23D, works, but both do not allow for the use of NOEs. Hence, a different software had to be used for the inclusion of NOEs in the second stage of WebCocon. A general search reveals many MD packages for small molecules, but most of them do not use NOEs and many of them have not seen updates for years [47]. A complementary search in Wikipedia [48,49] reveals several MD packages, most of them designed for biopolymers. From these, Tinker v8.8.3 [50] was identified as candidate, based on easiness of implementation and inclusion into the automation, as the Tinker molecule file format can be read and written by OpenBabel. Tinker also has a distance geometry (DG) module, which is much better suited for the generation of 3D coordinates starting with a connectivity list than MD, as it derives the coordinates directly from interatomic distances. With this, the inclusion of experimental distances such as NOEs into the structure calculation is easily performed, as they are included as interatomic distances. Since the quality of the DG results depends on the size of the set of generated structures, a short (90 structures) and a long (499 structures) version of the processing scripts were implemented. In both cases, the lowest energy structure from the set is chosen as the solution for a given constitution. The total energy of the conformation includes the contribution of the NOE violations, thus reflecting how well they were fulfilled.

WebCocon is available as a free-to-use service. It does not require registration and abstains from any tracking. All results discussed below are available for viewing on a dedicated page on the server.

Three molecules were selected to exemplify the results obtained (Figure 2). Caffeine (**1**) was chosen to discuss the question of reciprocity of molecules and correlation data, as the complete theoretical data set was experimentally observed. The marine natural product oxocyclostylidol (**2**) serves as an example for the use of ${}^{4}J$-HMBC correlation data because several identified experimentally observed ${}^{4}J$-HMBC correlations were available [51]. The diterpene pyrone asperginol A (**3**) was chosen as example for the use of NOE data in CASE because, besides good-quality NMR data, including 15 NOEs, a reference X-ray structure was available [52]. All NMR data available for the molecules **1**–**3** is summarized in Table 1.

## 2. Results

### 2.1. Reciprocity of Molecules and Correlation Data

It is generally accepted that NMR correlation data might fit more than one constitution, which justifies all CASE efforts. However, there is no measure of the ambiguity of NMR data for a given molecule. In order to address this question, WebCocon can generate a complete theoretical NMR correlation data set (COSY, HMBC, NHMBC, and ADEQ data) for a molecule. These data can then be submitted to the WebCocon server for a structure elucidation [32].

To illustrate this ambiguity, caffeine (**1**) was taken as example. The complete theoretical data set of **1** comprises eight HMBC and six NHMBC correlations (Table 1) and matches the experimental data set. Unlike reported for other purines [53], we did not observe long-range HMBC correlations. Additionally, all connections between two nitrogen atoms, or a nitrogen atom and an oxygen atom were forbidden. With this data set and restrictions, WebCocon still generates three structural proposals (Figure 3). This means that using the complete set of NMR correlations, a distinction between them is not possible. Structures **1**-1 and **1**-2 are difficult to distinguish by NMR correlations.

In order to come to a conclusion, ${}^{13}$C NMR chemical shifts were calculated for the structural proposals [36] using three different calculation methods: NMRShiftDB [54] (**M-I**), DFT (GAMESS 2019 R2 [55], **M-II**), and NMRPredict [56] (**M-III**). The results were compared to experimental values, as shown in Table 2. The data calculated from NMRShiftDB matches very well for **1**-1, with an overall average deviation of only 1.1 ppm. For **1**-2, NMRShiftDB issues a warning that the prediction quality is really bad and that matches with the overall average deviation of 23.5 ppm. Using DFT, we observe an overall average deviation of the chemical shifts of 2.8 ppm for **1**-1 and 8.3 ppm for **1**-2. The predictions by NMRPredict are slightly better, with overall average deviations of 2.8 ppm for **1**-1 and 7.3 ppm for **1**-2. Considering these values, **1**-1 would be chosen as the solution. Additionally, the chemical shift variations for positions 6 and 12 are significant enough for a distinction between **1**-1 and **1**-2.

While the back-calculated data matches very well for **1**-1, the back-calculated data for **1**-2 was marked by NMRShiftDB as very inaccurate. Similarly, the values obtained for **1**-2 by DFT do not match the experimental chemical shifts very well. However, still, the chemical shift variations for positions 8 and 12 are significant enough for a distinction between **1**-1 and **1**-2.

### 2.2. Use of ${}^{4}J_{\mathrm{CH}}$ Correlation Data

The cyclic monomeric pyrrole-imidazole alkaloid oxocyclostylidol (**2**) was chosen as an example for the structure elucidation with ${}^{4}J$-HMBC correlation data. Oxocyclostylidol (**2**, Figure 2) isolated from the Caribbean sponge *Stylissa caribica* was first published 15 years ago [51] and seems to be the perfect candidate for this investigation since four ${}^{4}J$-HMBC correlations were observed experimentally (besides 25 HMBC correlations, Table 1). The complete experimental data set of **2** is represented as data set **A** in Table 3. With this data set, WebCocon generated four possible solutions shown as **2**-1, **2**-2, **2**-3, and **2**-6 in Figure 4. These results were reproduced with the actual version of WebCocon.

The CASE investigations of oxocyclostylidol (**2**) were repeated using WebCocon with several different combinations of the experimental ${}^{4}J$-HMBC correlations, and the results are summarized in Table 3. The systematic investigation of the ${}^{4}J$-HMBC correlations of **2** started with the full data set (data set **A**) and without any ${}^{4}J$-HMBC correlations (**A0**, the letter stands for the data set and the number represents the 4J-Flag), which resulted in four structural proposals as we obtained before (Figure 4). The calculation time for the standard WebCocon run is less than one second. If all HMBC correlations were allowed to be two-, three-, or four-bond interactions (data set **A** with 4J-Flag = −1), the calculation time increases by a factor of 1000 (15 min and 7 s) and the number of solutions from 4 to 6045. This already clearly indicates that allowing all HMBC correlations to be a ${}^{4}J$ correlation is not a practical approach.

In the next step, we included only one of the ${}^{4}J$-HMBC correlations to the input data of the WebCocon calculations, which increased the number of HMBC correlations to 26 (data sets **B**–**E**). If we include the ${}^{4}J$-HMBC correlations and run WebCocon in the standard version (4J-Flag = 0), no solution is found, as expected. If we allow one of the 26 HMBC correlations (data sets **B**–**E**) to be a ${}^{4}J$ correlation (4J-Flag = 1), three of the four calculations resulted in four structural proposals (**B1**, **D1**, and **E1**). Since the data set of **2** is already very well defined, the one ${}^{4}J$ correlation does not improve the results anymore. The interesting point is that the number of solutions increases in one of the calculations (**C1**) from four to six (Figure 4). That is a surprise because the number of structural proposals is expected to stay the same or to be less than the reference data set (with one correlation less). This observation can only be explained by the fact that the actual ${}^{4}J$ correlation of these data was interpreted as ${}^{2}J$ or ${}^{3}J$ correlation and another HMBC correlation was interpreted as ${}^{4}J$ interaction. A closer inspection of the two new structural proposals confirms this hypothesis (Figure 4).

In the next steps, two (data set **F**), three (data set **G**), and four (data set **H**) of the ${}^{4}J$-HMBC correlations were added to the data set of **2**. If data set **F** is run with 4J-Flag set to 1 (**F1**), no solution is found. This is to be expected because two of the 27 HMBC correlations are ${}^{4}J$ correlations. The same is obtained for the data set **G** when the 4J-Flag is set to 1 or 2 (**G1**, **G2**) as well as, for the data set **H**, when the 4J-Flag is set to 1, 2, or 3 (**H1**, **H2**, and **H3**). In all cases, the number of experimental ${}^{4}J$ correlations is larger than the allowed ${}^{4}J$ correlations (4J-Flag) in the WebCocon calculations.

For data set **F** with 4J-Flag set to 2 (**F2**), for data set **G** with 4J-Flag set to 3 (**G3**), as well as for data set **H** with 4J-Flag set to 4 (**H4**), six structural proposals were obtained. In all cases, the ${}^{4}J$ correlation (from H-7 to C-3), which increased the number of solutions in the calculations with data set **C**, is included in these data sets. Several conclusions can be drawn from Table 3:Allowing ${}^{4}J$-HMBC correlations in the structural elucidation when there are none present in the input data increases the calculation time and possibly the number of results dramatically;The presence of ${}^{4}J$-HMBC correlations in the input data without allowing the ${}^{4}J$-HMBC interpretation during CASE makes the process fail;The best results are obtained when using no ${}^{4}J$-HMBC correlation data or when the number of allowed ${}^{4}J$-HMBC correlations in the CASE run matches the number of actually present ${}^{4}J$-HMBC correlations.

Interestingly, the four constitutions generated by WebCocon when using no ${}^{4}J$-HMBC correlations also are found when running calculations with one ${}^{4}J$-HMBC correlation. In the job that includes the H-7/C-3 ${}^{4}J$-HMBC correlation, a total of six solutions are generated, the four already known and two new ones, all shown in Figure 4. The results **2**-4 and **2**-5 were obtained because WebCocon could interpret the ${}^{4}J$-HMBC correlation as HMBC correlation and then change the interpretation for a HMBC correlation to ${}^{4}J$-HMBC.

### 2.3. Use of NOE Data in WebCocon’s Second-Stage Processing

The proton-rich diterpene pyrone asperginol A (**3**) was chosen for the application of WebCocon calculations using NOEs (Figure 5), because NMR and X-ray data were available, allowing for a comparison of the results [52]. The experimental data set comprises 18 COSY and 38 HMBC correlations (Table 1). Additionally, 15 NOEs were used in the structure discussion in the publication (Table 1). The 15 NOEs were defined as a range of 1.8 Å–4.0 Å for the use of WebCocon, as no individual quantification was available. In total, WebCocon generated 204 solutions, including different assignments, with 90 being unique constitutions. The default MD-based second-stage processing regards only the 90 unique constitutions, but processing including NOEs has to take all assignments into account and therefore takes considerably longer. The correct constitution was ranked around position 5 in different CASE runs, always using the same data. The better ranked constitutions exhibit varied substitution patterns in ring **A**, for which no NOEs were available.

WebCocon uses the force field total energy of the MD- or DG-generated conformation to rank the suggested constitution. The ranking for the correct constitution did not change significantly, when NOEs were introduced into the second-stage processing. However, superimposing the suggested conformations from MD processing, long MD processing, and DG processing to the available X-ray structure, shows that only the DG processed conformations are similar to the X-ray reference (Figure 6).

## 3. Discussion

The results shown clearly indicate that the fastest way to achieve a small set of suggested constitutions is the exclusion of ${}^{4}J$-HMBC correlations. Since this is not always possible, the best strategy seems to be a step-by-step increase of the allowed ${}^{4}J$-HMBC correlations until a set of suggestions is obtained. This process shall be automated in the future.

The use of NOE data in the second-stage processing improved the quality of the conformation suggested as the solution when compared to the crystal structure. However, this did not change the ranking of the correct conformation, as alternative structures fit the experimental data equally well. This can be due to the choice of NOEs used (only NOEs provided by the authors were used), due to the fact that all NOEs were defined with the same distance range, or due to the lack of explicit protons used. For the future, the inclusion of more NOEs and the better definition of their distances (e.g., characterized as strong, medium, and weak) can lead to better results. Furthermore, a method of using explicit protons for the definition of NOEs is being developed. This is a first step bringing automated constitutional analysis and automated configurational/conformational analysis together.

All of this automation becomes of special interest when combined with initiatives such as NMReDATA [57,58,59], which allow for easy and comprehensive data exchange of all spectroscopic data associated with a molecule. WebCocon can read the parts of this format that are relevant for the generation of all inputs needed for a comprehensive structure discussion using experimental data.

## 4. Conclusions

Our continued interest in the development of CASE systems has led us to further improve the web-based CASE software WebCocon. As new feature, the software is now capable of using ${}^{4}J$_CH_ HMBC and NOE correlations. There are not many examples reported in the literature for either case. Of general importance is the underlying question, to which extent such CASE systems could be helpful to researchers in the real world. As initial examples we calculated all constitutions compatible with the 2D NMR data sets of the marine natural product oxocyclostylidol (**2**) and the diterpene pyrone asperginol A (**3**), and their molecular formulae. The structurally simple example caffeine (**1**) was included to highlight an already existing feature of WebCocon that is considered very important whenever a structural proposal is to be analyzed for the existence of alternatives. Indeed, there is even an alternative to caffeine.

Since it is never known, which of the experimentally observed HMBC correlations have to be translated to a connectivity over four bonds, a certain percentage of those is to be declared as ${}^{4}J$_CH_ correlations, stepwise. For oxocyclostylidol, we went up to about 20% and still were able to obtain a manageable number of constitutions. In reality, oxocyclostylidol exhibits four ${}^{4}J$_CH_ correlations. There is experimental evidence that many of the investigated compounds in the literature have at least one HMBC correlation over four bonds. In this case, every standard automated structural elucidation would fail because this correlation could not be correctly translated.

The inclusion of distance information (through NOEs or ROEs) as demonstrated here is the first step towards the generation of real conformations of small molecules as a result of the NMR data interpretation. In the end, with this approach, not only structure elucidation but also a reliable configuration and conformation determination can be achieved starting with a full NMR data set that could be contained in a NMReDATA archive.

## Figures and Tables

**Figure 1 molecules-26-04846-f001:**
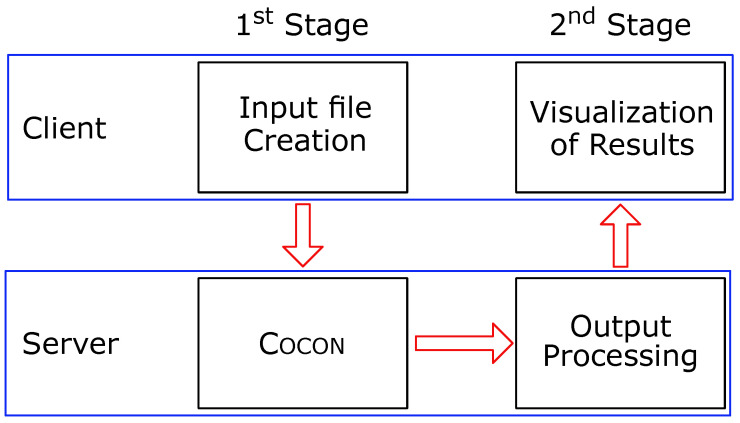
WebCocon uses a two-stage workflow. The first stage begins with the input file creation (on the client) followed by the Cocon run, which generates a list of connectivity sets, each set representing one constitution. In the second stage, this set of connectivities is converted into 2D/3D molecular information ranking the candidates that can be visualized on the client. The second stage can be repeated using any of the (currently four) processing methods available.

**Figure 2 molecules-26-04846-f002:**
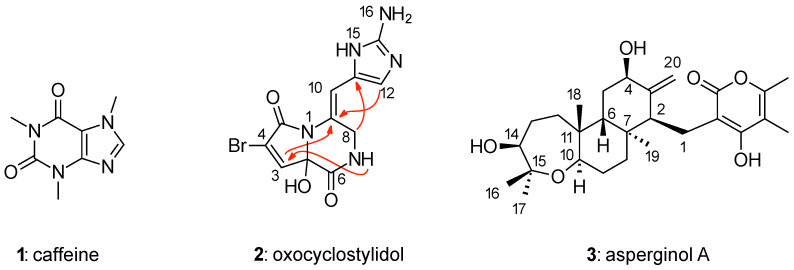
Structures of the investigated molecules **1**–**3**. For oxocyclostylidol (**2**) the observed HMBC correlations over four bonds are indicated as red arrows.

**Figure 3 molecules-26-04846-f003:**
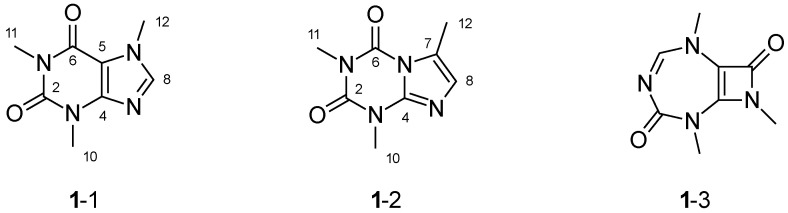
Based on the theoretical NMR correlation data set for **1**, WebCocon generates the two alternative constitutions **1**-2 and **1**-3.

**Figure 4 molecules-26-04846-f004:**
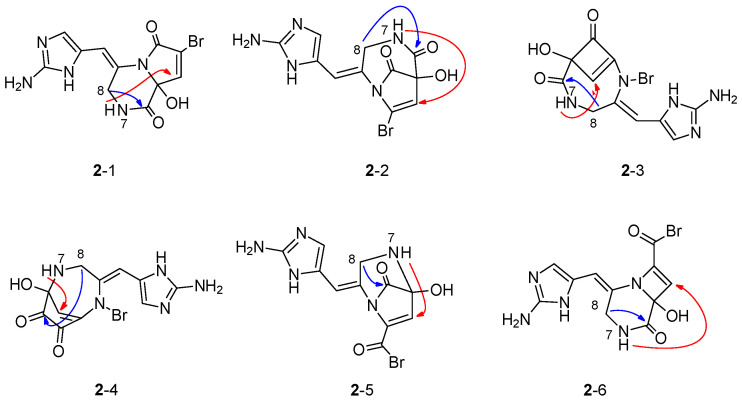
Constitutional proposals for oxocyclostylidol (**2**) generated by WebCocon. For the data set without ${}^{4}J$ correlations (**A0**) and three data sets with one ${}^{4}J$ correlation (**B1**, **D1**, and **E1**), four constitutions were found (**2**-1, **2**-2, **2**-3, and **2**-6); for data set **C1**, all six structures were generated. In the proposals **2**-4 and **2**-5, the ${}^{4}J$-HMBC correlation H-7/C-3 (red arrows) was fulfilled as HMBC correlation and the HMBC correlation H-8/C-6 (blue arrows) was interpreted as ${}^{4}J$-HMBC correlation.

**Figure 5 molecules-26-04846-f005:**
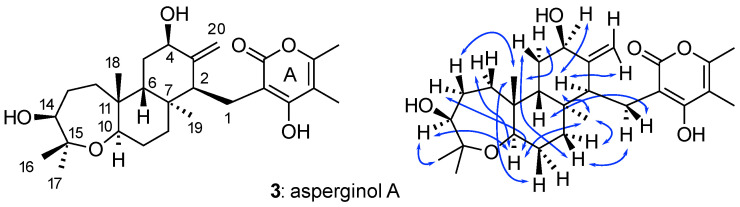
Asperginol A (**3**) and the 15 NOEs (in blue) included in the structural elucidation.

**Figure 6 molecules-26-04846-f006:**
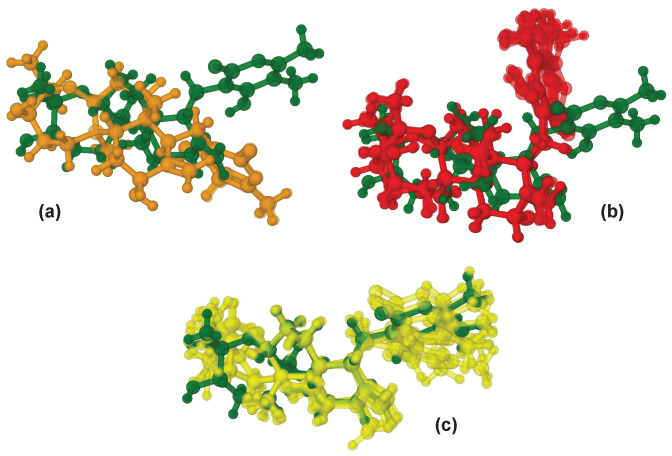
Superposition of the crystal structure of **3** (green) with the five best conformations obtained by (**a**) MD (orange), (**b**) long MD (red), and (**c**) DG with NOEs (yellow).

**Table 1 molecules-26-04846-t001:** Correlation data (number of correlations) of the investigated molecules **1**–**3**.

	Data	COSY	HMBC	${}^{4}J$-HMBC	ADEQ	NHMBC	NOE
Caffeine (**1**)	theo. ^a^	–	8	–	–	5	–
Oxocyclostylidol (**2**)	exp.	1	25	4	6	9	–
Asperginol A (**3**)	exp.	18	38	–	–	–	15

^a^ The experimental data set of **1** is identical to the theoretical data set.

**Table 2 molecules-26-04846-t002:** ${}^{13}$C NMR chemical shifts [ppm] for caffeine (**1**-1) and the imidazotriazine (**1**-2), including the average deviation $\left(\bar{\Delta }\right)$ to the experimental values for each of the calculation methods.

		1-1			1-2		
**Atom**	**exp.**	**M-I** **^a^**	**M-II** **^b^**	**M-III** **^c^**	**M-I****^a^**⚠	**M-II** **^b^**	**M-III** **^c^**
2	148.5	150.7	149.5	151.4	151.1	152.4	150.8
4	151.5	149.0	153.3	147.0	157.4	152.0	149.7
5	107.4	107.3	104.5	111.5			
6	155.2	155.3	154.7	154.3	149.2	149.0	149.4
7					60.1	115.9	117.1
8	141.4	143.0	145.8	147.4	50.8	122.3	128.2
10	27.8	28.8	25.7	29.5	37.3	27.1	31.3
11	29.6	28.7	26.7	27.7	37.3	27.2	29.3
12	33.5	33.4	26.9	33.7	15.4	7.9	11.6
$\bar{\Delta }$		1.06	2.78	2.78	23.46	8.25	7.31

^a^ Calculated by NMRShiftDB, “⚠” means the values are not reliable. ^b^ Calculated by DFT (GAMESS 2019 R2). ^c^ Calculated by NMRPredict.

**Table 3 molecules-26-04846-t003:** Number of solutions generated by WebCocon, depending on the ${}^{4}J$ correlations included in the data set, number of allowed ${}^{4}J$ correlations in structure generation, and computer time used (averaged over three runs, on an Intel Core i7-3770 processor system).

Input		${}^{4}J$-HMBC				Cocon	
**Data Set**	**4J-Flag**	**H-3/C-9**	**H-7/C-3**	**H-8/C-11**	**H-12/C-9**	**sol.**	**Run Time [s]**
**A**	0	-	-	-	-	4	1
	1	-	-	-	-	18	30
	2	-	-	-	-	107	42
	3	-	-	-	-	329	76
	4	-	-	-	-	889	153
	−1	-	-	-	-	6045	907
**B**	0	X	-	-	-	0	0
	1	X	-	-	-	4	17
	2	X	-	-	-	19	20
	3	X	-	-	-	116	33
	4	X	-	-	-	330	66
	−1	X	-	-	-	3974	525
**C**	0	-	X	-	-	0	0
	1	-	X	-	-	6	23
	2	-	X	-	-	32	27
	3	-	X	-	-	167	46
	4	-	X	-	-	529	98
	−1	-	X	-	-	4664	592
**D**	0	-	-	X	-	0	0
	1	-	-	X	-	4	27
	2	-	-	X	-	18	30
	3	-	-	X	-	107	42
	4	-	-	X	-	329	74
	−1	-	-	X	-	6045	788
**E**	0	-	-	-	X	0	0
	1	-	-	-	X	4	28
	2	-	-	-	X	18	31
	3	-	-	-	X	108	43
	4	-	-	-	X	346	79
	−1	-	-	-	X	6045	791
**F**	0	X	X	-	-	0	0
	1	X	X	-	-	0	13
	2	X	X	-	-	6	14
	3	X	X	-	-	31	19
	4	X	X	-	-	172	39
	−1	X	X	-	-	2910	402
**G**	0	X	X	X	-	0	0
	1	X	X	X	-	0	14
	2	X	X	X	-	0	14
	3	X	X	X	-	6	15
	4	X	X	X	-	31	18
	−1	X	X	X	-	2910	400
**H**	0	X	X	X	X	0	0
	1	X	X	X	X	0	14
	2	X	X	X	X	0	14
	3	X	X	X	X	0	14
	4	X	X	X	X	6	14
	−1	X	X	X	X	2910	401

## Data Availability

All results shown in this article can be visualized by accessing the corresponding page on the WebCocon Server: https://cocon-nmr.de/publication_data (accessed on 25 June 2021).

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
