# Peer review of "Incorporation of ^4^*J*-HMBC and NOE Data into Computer-Assisted Structure Elucidation with WebCocon"

_molecules, 2021, doi:10.3390/molecules26164846_

Round 1

Reviewer 1 Report

The paper describes a potentially useful extension of a web server for NMR-based small molecule structure elucidation. The paper is well written. I recommend publication after addressing the following points:

1. Lines 49-54: I cannot see any fundamental reason for not referring NOEs to hydrogen atoms instead of heavy atoms. This seems to be a purely technical limitation of the software that can lifted easily.
2. Lines 58-60: Fundamentally, MD is a much better method than distance geometry (DG) for structure calculation. Multiple MD packages are available that can handle NOE distance restraints.
3. Line 156: also are —> are also
4. Line 166: use decimal points instead of commas.
5. Line 174-175: This might well be different for other molecules and/or with other NOE data.
6. Table 1: The number of three test examples is minimal. In particular, using NOEs was tested (unsuccessfully) only for a single molecule. 
7. Table 3: Give the name of the molecule in the Table heading.
8. Lines: 161-164: This procedure appears to be cumbersome. Spending a bit more computation time in a simpler procedure might better for most users.
9. Line 191: This is almost trivial to do and likely to help.
10. Line 192: This may indeed make a significant difference.

Reviewer 2 Report

1 ) No references are provided for other efforts to develop CASE programs. At least, the extensive work by Mikhail Elyashberg should be cited since this issues is in his honour. One easy way to correct this would be by citing a recent comprehensive review of CASE programs by Elyashberg and others that appeared earlier this year in Magnetic Resonance in Chemistry (working at home, I don't have the exact reference available).

2) Part 1, compound 1: This is an interesting idea, but one point needs clarification. My inspection of the 3 structures indicates that one of the six NHMBC correlations (involving the methine proton) is a 4-bond correlation in each case. Does WebCocon routinely allow 4-bond NHMBC peaks?

3) It is interesting that relatively few structures fit the complete set of possible correlations for compound 1. Experimentally, it is rare to have a complete set of HMBC data in particular. For future work, it might be informative to compare results where all possible correlations for a compound are included with those where a fraction (10-30%) were omitted. I suspect that the latter runs would generate more alternative structures. If so, this would highlight the importance of obtaining the best possible data in a structure elucidation to minimize the risk of proposing a wrong structure.

2) The ability to handle 4-bond HMBC peaks is an important Improvement for WebCocon. However, particularly in planar conjugated systems, 5-bond HMBC peaks can also be observed. In fact, in the original report for compound 2 (ref. 14), a 5-bond peak (H-12/C-8) was reported. The authors should perform similar runs to those in Table 4, where this peak is included, along with 'regular' HMBC peaks, in order to see what impact that had on structure generation. My guess is that no structures would be generated, indicating the need to inspect the data to exclude very weak peaks.

4) If future improvements of WebCocon are undertaken, I would suggest altering the option of including different numbers of 4-bond peaks durng structure generation to instead allow for any peaks 4-bonds or longer. I believe that this option is available in some other CASE programs, including one developed mainly by Mikhail Elyashberg. Alternatively, if allowing 4-bond peaks failed to generate a structure, only then could longer range correlations be allowed. Doing this in two stages should minimize the risk of generating excessive numbers of possible structures in cases where no HMBC correlations for longer than 4 bonds were present.

5) Part 3, structure 3: This is a useful first attempt to tackle the difficult problem of generating 3D structures from NMR data. As the authors indicate, more work will be needed in the future to better evaluate the utility of this approach by including at least semi-quantitative nOe data  for target molecules 

Reviewer 3 Report

In this manuscript, Junker et al. report on incorporating four-bond HMBC correlations and NOEs into CASE. In general, long-range couplings and NOE contacts are very useful in structure elucidation and their implementation to the CoCon tool is really important. The authors present three molecules as examples how the additional parameters help in structure determination and which difficulties must be overcome. However, in my point of view, there are still some points, which should be clarified before publishing.

  • Three structurally diverse molecules are present to demonstrate how useful the method is. Are the three molecules sufficient for verification?
  • Introductory part is lacking. I would suggest to extend introduction, add two paragraphs and write more about 1) long-range interactions and 2) NOE, stress their importance in structural analysis and show some structurally diverse examples, where these NMR parameters are essential. For this purpose, the authors should also provide more citations, e.g. when talking about caffeine, I would add a reference about long-range couplings in purines ( Magn. Reson. 2012, 50, 295-298) etc.
  • Table 1 is not consistent, theoretical and experimental approaches are mixed. I do not understand, why the authors show only theoretical data for caffeine. In my opinion, they should measure this compound and provide all the parameters to fulfill Table 1.
  • In table 1, the authors say that caffeine has no 4J HMBC. However, H6-C3 contact should be considered and, may be the crucial parameter able to distinguish 1-1 and 1-2 without 13C calculations. If the 4J HMBC can distinguish them unambiguously, I would exclude 13C calculations and also Table 2 from the manuscript. Then, the proposed method improvements will be stressed even more.
  • The authors want to implement information about HMBC correlation and not extract the coupling constants. I do understand that for screening purposes it is much more convenient. However, how do the authors distinguish between the “standard HMBC correlations” (probably 3J and 2J, p. 3, line 107) and the others (long-range – 4J and 5J)? Is it possible to somehow define in the program, which crosspeaks correspond to 3J and which are long-range?
  • How do the authors distinguish which crosspeaks correspond to four-bond couplings? The crosspeak intensity in HMBC strongly depends on the parameters set in the pulse-program. Then, 4J and 5J can be misinterpreted. Then, is 4J-HMBC really a correct nomenclature?
  • I do not understand Table 3 and what is the take-home message. The caption should be self-explanatory, including information about the compound described and legend. May be, a short paragraph summarizing the table would be helpful.
  • I would avoid using the abbreviations cm-1, cm-2 and cm-3 for calculation methods, because it evokes reciprocal centimeters to me. Why not to use just simple I-III? The three cm methods should be mentioned in section Materials and Methods if the caffeine calculations will not be excluded or substituted by the experimental data I already proposed above.
  • In the article title and in abstract (line 1) and throughout the manuscript. Explanation of the CASE abbreviation should be corrected. CASE is published as Computer-Assisted Structure Elucidation ( Chem. Educ. 2020, 97, 3, 855–860).
  • In abstract (line 4), “integration of 4J-HMBC correlation”: This is confusing. I would change the word integration into implementation, because integration has a different meaning in NMR.
  • All the chemical structures should be the same size and definitely much smaller than that in Figure 1.
  • Structure of 2-1 is duplicated in Figure 2 and 3. I would merge these two into one Figure.
  • Figures 6-8 show the same molecule treated by different ways, thus, can be merged into one figure – Figure 6 a, b, c.
  • Decimal separator should be a point, not comma throughout the manuscript.
  • I did not find the section Conclusion.
  • 9, line 185: … “did not improvement” should be corrected to “did not improve”
  • 9, line 192: …”is been developed” should be corrected to “has been developed “OR “is being developed.” Language corrections would improve the manuscript.
  • References are not formatted in a consistent way, e.g. ref. 14. Abbreviations of the journal names cited will be sufficient. Interestingly, only 4 references from 18 are newer (published after 2018). The reference update may be appreciated.
  • Abbreviations:

CASE – Computer Aided Structure ELucidation should be corrected to Computer Assisted Structure Elucidation (including the typo in ELucidation)

DFG – Density functional theory should be corrected to DFT

Round 2

Reviewer 3 Report

I thank the authors for their responses and I recommend to accept this article in its present form.